# A Systematic Review over the Effect of Early Infant Diet on Neurodevelopment: Insights from Neuroimaging

**DOI:** 10.3390/nu16111703

**Published:** 2024-05-30

**Authors:** Dylan Gilbreath, Darcy Hagood, Linda Larson-Prior

**Affiliations:** 1Department of Neurobiology and Developmental Sciences, University of Arkansas for Medical Science, Little Rock, AR 72207, USA; ljlarsonprior@uams.edu; 2Arkansas Children’s Nutrition Center, Little Rock, AR 72202, USA; hagooddr@archildrens.org

**Keywords:** infant diet, neurodevelopment, neuroimaging, infant nutrition, human milk, magnetic resonance imaging, electroencephalography

## Abstract

The optimization of infant neuronal development through nutrition is an increasingly studied area. While human milk consumption during infancy is thought to give a slight cognitive advantage throughout early childhood in comparison to commercial formula, the biological underpinnings of this process are less well-known and debated in the literature. This systematic review seeks to quantitatively analyze whether early diet affects infant neurodevelopment as measured by various neuroimaging modalities and techniques. Results presented suggest that human milk does have a slight positive impact on the structural development of the infant brain—and that this impact is larger in preterm infants. Other diets with distinct macronutrient compositions were also considered, although these had more conflicting results.

## 1. Introduction

The structural and functional development of the human brain has yet to be fully elucidated, and the effect of infant diet on neuronal development is becoming an increasingly studied area. Seminal research on developmental neuropathology suggests that environmental influences—including nutrition—have the greatest impact on the development of the brain’s structure [1], one that endures in subsequent development of various neuronal processes. In this way, infant diet has a persistent effect on the structure of the brain. While many studies retrospectively link early infant diet to developmental outcomes in older children and adolescents, relatively few studies have been conducted on the impact of diet on the structure and function of the infant brain during its development. The purpose of this review is to thoroughly examine the influence of human milk or various formula infant diets on neurodevelopment measured quantitatively through neuroimaging.

Human milk as a primary nutrient source for the first 6 months of life, and until 2 years of age with complementary feeding, is the current diet recommended by the American Academy of Pediatrics for optimal development [2]. This is supported by an extensive literature demonstrating that breast-fed (BF) infants have a slight cognitive advantage quantified by various intelligence measures over their formula-fed (FF) counterparts, and that this effect persists throughout childhood and adolescence [3,4,5,6,7,8,9]. This effect on cognition is independent of maternal education and intelligence scores [10], has been replicated cross-culturally [7,9], and is often used to support the claim that human milk is the optimal diet for infant development. However, the way in which the underlying neuronal structures are developing and subsequently functioning under the influence of these different diets to account for this cognitive difference is still unclear.

During infancy, the brain undergoes rapid neurodevelopment fueled by adequate nutrition, which provides critical components to build the nervous system [11]. To determine what these critical components are, many studies have focused on specific compounds inherent in human milk, with human milk oligosaccharides (HMOs) and polyunsaturated fatty acids (PUFAs), including docosahexaenoic acid (DHA), being the most commonly researched. PUFAs have been identified as important for neurodevelopment [12] and are an essential nutrient in human milk, since synthesis of certain fatty acids in infancy is constrained by the limited enzyme activity of fatty acid desaturase (FADS) [13,14]. DHA is of particular interest because of its hypothesized importance for neurogenesis [15], and its high concentration in the frontal lobe [16]—an area heavily associated with executive function. Because of this, DHA is commonly added to commercial formulas [13], and some studies have found that DHA supplementation in formula increases cognitive scores later in life [17,18]. However, other studies report that DHA supplementation of commercial formula results in no improvements of overall cognitive scores [19,20,21], while one study found that DHA supplementation without the addition of arachidonic acid (AA) did not have an effect on cognition [22]. These discrepancies may arise from differences in the age of subjects at the time of assessment, additional supplementation with other PUFAs, or biologically from individual differences in FADS genotypes that are manifesting at a population level [13]. These conflicting studies demonstrate a gap in understanding of which components of human milk are related to these differences, and whether these differences are manifesting in the structure or function of the developing brain.

While behavioral studies examining overall markers of intelligence present clear evidence that human milk does subtly enhance cognition and neuromaturation, the exact components responsible for this change is an active field of research. However, in the absence of neuroimaging studies showing differences in the actual structure and function of the developing brain, conclusions on dietary effects on neurodevelopment should be drawn cautiously. Advances in neuroimaging equipment and processing methods have made it possible to evaluate brain development in infants. The brain’s physical structure can be measured using magnetic resonance imaging (MRI), including diffusion tensor imaging (DTI) and functional MRI (fMRI) as modalities. Structural MRI methods are often chosen in infant studies because of the known, dramatic developmental changes in the brain’s structure including neuronal proliferation and migration, myelination, and synaptogenesis [23,24]. MRI can also evaluate the functional architecture of the developing infant brain indirectly through the blood oxygenation level-dependent (BOLD) signal [25]. While MRI is known for its high structural resolution, it lacks temporal resolution in comparison to other methodologies [26]. Neuronal function using electroencephalography (EEG) is increasingly used to determine infant neurodevelopment since EEG directly measures neuronal function in the millisecond range by detecting changes in scalp current densities produced by cortical pyramidal neurons [27]. These modalities are complementary– functional connectivity measured by EEG in the first months of life mirrors the synaptogenesis and myelination measured by MRI that is known to occur during this time frame [28]. By including studies examining both the structure and function of the developing brain, the impact of nutrition can be better quantified and assessed.

This systematic review seeks to present evidence from the neuroimaging literature exploring early infant diet’s effect on the structure and function of the brain during the timeframe in which the brain is undergoing both rapid development and likely still obtaining nutrients from either human milk or formula (from birth to two years).

## 2. Methods

The PRISMA (Preferred Reporting Items for Systematic Reviews and Meta-Analyses) statement checklist was used to determine both eligibility for this study and how search metrics are reported [29].

### 2.1. Search Strategy

All eligibility criteria and methodologies were developed a priori; these strategies were limited to the English language, and included terms related to both infant nutrition and various neuroimaging techniques. PubMed and ScienceDirect were the electronic databases used to search for relevant papers, and papers selected were manually reviewed for additional citations. All results from search strategies were manually tracked and recorded. Searches were not explicitly restricted by year published, but interest in infant nutrition with respect to neuronal development was rarely studied until the early 2000s (Figure 1). The search strategy involved a combination of infant-diet-and neuroimaging-related terms; “Breast milk” OR “Human Milk” OR “Infant Diet” OR “Infant Nutrition” AND “Neuroimaging” OR “MRI” OR “EEG” OR “MEG” OR “DTI”. The full search terms used in the PubMed database can be found in Figure A1.

### 2.2. Inclusion and Exclusion Criteria

Inclusion and exclusion criteria are detailed in Table 1 using the PICOS format [30]. Studies examining only healthy infants were included; therefore, studies concerning either specific nutritional deficiencies (see [4] for a review) or studies analyzing specific diseases or developmental disorders were excluded. Preterm infants were included due to the potential for diet to have a differential effect on development depending on when a diet or intervention was introduced. Studies involving preterm infants were included if the infants were otherwise healthy (no known congenital abnormalities, illnesses, comorbidities, or expected deviations from normal development) and met all other inclusion criteria. Diets looking at the specific supplementation of a nutrient (L-PUFAS, additionally fatty acids, etc.) in an otherwise healthy diet were included. Reviews, abstracts, studies that were not yet complete, study protocols, and studies that exclusively looked at older children were all considered ineligible. While magnetoencephalography (MEG) was considered as a viable neuroimaging technique since it provides a non-invasive way to measure neuronal activity with a millisecond temporal resolution, no studies were uncovered by our literature search despite its increasing use in infants [31]. Neuroimaging techniques including or encompassing EEG and MRI were included. One reviewer (DG) screened all titles and abstracts of the identified studies from the initial search and determined a preliminary eligibility status. A second reviewer (LLP) independently reviewed the 72 preliminary articles selected for a full-text review and arrived at a consensus with the first author on which papers should be included.

### 2.3. Data Extraction

Data were extracted in a standardized form including primary author, publication date, age and total number of subjects, type of diet/nutritional intervention, neuroimaging modality, study paradigm, and primary findings.

## 3. Results

### 3.1. Selection of Studies

Following PRISMA guidelines, electronic search of selected databases yielded 4704 articles for review; after duplicates were removed and our exclusion criteria added, 2528 articles remained. After an initial screening reviewing abstracts and titles, 72 were selected for a full text review of which 45 were excluded for not meeting eligibility criteria. Eight articles were manually selected from references in relevant papers, bringing the total number of papers selected for this review to 27. These results are summarized in Figure 2.

### 3.2. Description of Studies

Table 2 summarizes the characteristics of the studies in this review. In summary, eleven studies used EEG (two preterm [32,33], nine term-born [34,35,36,37,38,39,40,41,42]) and sixteen used MRI (eleven preterm [43,44,45,46,47,48,49,50,51,52], five term-born [3,53,54,55,56]) as a modality. To evaluate diet’s effect on neuronal function at a specific developmental time point using EEG, eight studies [32,33,34,35,36,38,39,41] used task-related evoked related potentials (ERPs)—a time-locked waveform that occurs in response to a stimulus—and three [37,40,42] used resting state analysis to measure spontaneous neuronal activity. Most MRI studies evaluated the structure of the brain in some way [44,45,46,47,48,49,50,51,52,53,54,55,56,57] while two studies [43,49] evaluated the function of the brain. To better image the overall structure of the developing brain, several studies used additional MRI techniques such as multicomponent driven equilibrium single pulse observation of T1 and T2 (mcDESPOT) that images myelin content [44,47,54,56] and diffusion tensor imaging (DTI) that images white matter tracts [3,50,52,53,56,57]. In terms of dietary effects on neurodevelopment, thirteen studies assessed the effect of human milk in comparison to formula, two studies examined the effect of the duration of human milk feeding, four studies assessed the macronutrient composition, one study assessed human milk oligosaccharides, and seven examined the effects of enhancing a specific nutrient source.

### 3.3. Study Quality

The quality criteria checklist (QCC) included in the Academy of Nutrition and Dietic manual as a recommendation for the systematic review process was used to evaluate the risk of bias and to evaluate the overall quality of each research article included [58]. To determine an overarching impression of the quality of studies included in a review, the QCC asks a series of yes/no questions about the overall study design, statistical methods used, interpretations of data, and the risk of bias. Using the QCC, three studies had a moderate risk of bias [34,36,44], while the remaining studies had an overall low risk of bias as indicated by Table 3. Three studies did not report their sources of funding [32,44,46], and two studies from the same author had a potential conflict of interest [54,56]. Four EEG studies did not report controlling for Type 2 family-wise error in their statistical methods [33,34,38,39]. No studies included had a high risk of bias.

### 3.4. Study Participants

The study participants were comprised of preterm and term born infants who had a neuroimaging measure between birth and 2 years of age. Term-born infants did not have any preexisting conditions, childhood diseases, or congenital abnormalities. While preterm infants are more susceptible to various complications, infants in these studies were otherwise healthy with no major complications at the time of the study and were not subsequently expected to deviate from normal development. In addition, studies conducted in preterm infants overwhelmingly had a neuroimaging measure conducted at term equivalent age (TEA), providing important early developmental data.

## 4. Discussion

### 4.1. EEG

#### 4.1.1. Long Chain Fatty Acids’ Effect on Term Infants

The literature search identified 10 relevant papers using EEG methods, of which two involved analysis of preterm infants. Studies conducted in infants born at term were confined to acquisition and analysis of the auditory oddball syllable discrimination and the resting state. Beginning as early as 28 weeks’ gestation, infants exhibit behavioral responses to sound [59]. This perception builds into the ability to discriminate between auditory stimuli (often different phonemes when focused on language) and is a critical component of language acquisition [60]. Syllable discrimination typically occurs between 3 and 6 months of age [61] and can be detected by analysis of ERPs that are time-locked to a stimulus onset. These ERPs have specific characteristic component waveforms that have been mapped onto how infants process language in early development [62], with the most relevant being the positive-going P1 component reflecting stimulus detection, the negative-going N2 component reflecting encoding of acoustic features [63], and the P350 component which is thought to be involved in phoneme priming and occurs in response to an unexpected stimulus [64,65]. The auditory oddball paradigm utilized in these studies audibly presents a frequent standard syllable (such as “pa”) with a less frequent deviant syllable (such as “ba”). Differences in ERP components between these syllables are thought to be indicative of auditory discrimination between phonemes, a hallmark of the burgeoning language system. To see if early infant diet had any effect on language development, six studies examined whether being BF or FF had any effect on auditory oddball-related ERPs.

Two studies examining the differences in auditory oddball ERPs between infants who were exclusively BF or fed a dairy-based formula (MF) for the first 4 months of life found no significant differences between groups in 3- and 6-month-olds [34,35]. An important consideration is that these two studies used the same population (Beginnings Study, ID#: NCT00616395 at clinicaltrials.gov) with similar sample sizes. The remaining four studies did report differences when accounting for an additional dietary group that were fed a soy-based formula (SF). In brief, Li et al. [36] found that BF infants at 3 months of age had greater P350 amplitudes in frontal regions when compared to both MF and SF infants, longer N250 and P350 latencies than SF, and no group differences related to diet at 6 months. However, these reported results are self-conflicting. The author notes that greater response amplitudes correlating to frontal brain regions could indicate greater neuromaturation in the BF infants, while also suggesting that the reported longer latencies could be related to a delay in speech processing for the BF infants. This discrepancy could be due to the non-standard method of analysis implemented in this study: the paper reports only using ERP data from the standard syllable while the standard methodology includes looking at the difference wave of the deviant ERP from the standard syllable ERP. Results should be interpreted carefully considering these methodological deviations. In a follow-up study using comparisons to the deviant syllable, Pivik and colleagues [38] did not replicate these results, finding no significant differences between groups at 3 months of age, and at 6 months of age finding a significantly lower P350 amplitude to the standard syllable in BF infants compared to both MF and SF groups. A more recent analysis in this same study population that included the additional covariates of infant sex and weeks spent in gestation did not replicate this finding in the 6-month-olds [41]. However, this study did find differences in regional latency at 12 months of age with SF infants having a longer P2 latency in a right temporal region of interest (ROI) than MF and a shorter P2 latency in a frontal left ROI than BF and MF, suggesting that the SF groups may be interpreting the speech sounds as non-speech stimuli [41]. Another study using this cohort examined 4- and 5-month-olds, reporting differences in P170 and P350 between BF and FF infants suggesting that BF infants may develop certain language acquisition processes earlier [39]. Taken together, human milk may have a slight positive effect on language acquisition; however, this effect is subtle and reported results have been inconsistent within the same study population.

Resting state EEG is used as a baseline assessment of neuronal activity, and the development of spectral power is thought to mirror the underlying maturation of cortical networks [66,67]. As such, many studies choose to examine power spectral densities (PSDs) which reflect the frequency content of the EEG signal that is known to be actively developing during infancy [68]. Three studies, using similar methodologies measuring the same dietary effects (BF vs. MF vs. SF), reported that BF infants had more power in higher frequency bands than their SF counterparts [34,40,42]. Regional differences were found in gamma (study-defined as 30–50 Hz), with BF infants having higher gamma in the left hemisphere than MF and SF counterparts [42]. Increases in high frequency bands such as beta and gamma are positively correlated with age and are generally associated with better cognitive processing [68]. In conjunction with this, SF and MF infants were observed to have a shift in power towards the lower frequency range at 6 months, with greater power than BF infants in the 0–3 Hz and 6–9 Hz bands [37]. During maturation, the power in these low frequency bands is known to decrease until adolescence where it resembles the adult frequency content [68,69]. Power increasing in higher and decreasing in lower frequency bands in the BF infants are potential markers of greater neuromaturation.

#### 4.1.2. Long Chain Fatty Acids’ Effect on Preterm Infants

Studies using EEG to examine the intersection between diet and neurodevelopment in preterm infants were scarcer and largely focused on supplementation of long chain polyunsaturated fatty acids (LCPUFAs) vs. supplementation using other fatty acids. Bouglé et al. [32] analyzed whether formula supplementation with LCPUFAs or with short chain PUFAs had any effect on auditory or visual ERPs at term equivalent age (TEA) and reported no differences between groups. Fatty acid intake’s effect on memory was assessed by Henriksen et al. [33] through either adding additional LCPUFAs—DHA and AA—to human milk or supplementing human milk with a control oil containing a mixture of soy oil and medium-chain triglyceride oil. This study found that infants receiving additional DHA and AA had smaller amplitudes to repetitions of a standard image during a memory-related ERP at 6 months [33]. This decrease in amplitude potentially reflects a greater instance of memory recall in the infants who received the supplemented human milk as recognition of a standard image is correlated with a decrease in ERP amplitudes. While DHA is a LCPUFA, differences in these studies may be accounted for by the different time points at which the EEG was acquired (TEA vs. 6 months old)—longer exposure to LCPUFAs may be needed to have a positive effect on neurodevelopment, or this effect may be constrained to neuronal processes related to memory.

### 4.2. MRI

#### 4.2.1. HMOs’ Effect on Term Infants

MRI studies involving both full-term and preterm infants suggest that human milk improves structural brain development [3,43,49,51,52,53,56,57]. In a longitudinal study of term infants from 10 months to 4 years, infants who were fed an exclusively human milk diet had an increase in white matter microstructural development in frontal and temporal regions [53]. Greater myelination is a marker of development [70], and frontal regions are associated with executive functioning in infants [71,72] and adults [73]. Another study in infants from 3 months to 5 years found that BF infants had more rapid development of myelin from ~1.5 to 2 years, and that this trajectory ultimately resulted in an overall increase in myelin at a 2-year time point as measured by mcDESPOT [3]. One potential explanation for these differences in BF and FF infants is the inherent exposure to human milk oligosaccharides (HMOs) that have previously been identified as important for cognition [74,75]. Exposure to specific types of secreted HMOs at 1 month of age is associated with differences in tissue microstructure, with a greater exposure to 3′-sialyllactose and 3-fucosyllactose being positively associated with greater fractional anisotropy values in regions of the brain known to be developing at this time [76]. These positive values are associated with an increase in myelination which may indicate an increase in structural connectivity [77]. While not the focus of the present literature review, these results are supported by studies conducted in older children who were exclusively breast-fed that found increases in white matter volume [78] and better functional development of gray matter [79] than their formula-fed counterparts. Taken together, a tentative conclusion can be drawn that these structural differences arise early in infancy and persist throughout childhood as a result of these early organizational changes.

#### 4.2.2. HMOs’ Effect on Preterm Infants

These results are replicated in the preterm literature, with human milk optimizing overall neurodevelopment [43,45,49,51,52,57]. In Niu et al., functional brain network architecture in preterm infants who were BF or FF were analyzed for the first-time using fMRI. While both groups were found to have small-world topologies characterized by high local clustering and short paths between different nodes or brain regions, BF infants had higher temporal global efficiency than FF groups. This higher efficiency is thought to represent better coordination between brain regions [80]. Specific brain regions have both better coordination and activation as reported by another fMRI study that found increases in brain activation in the right temporal lobe of BF infants as well as increases in grey matter volume in the frontal lobes, right temporal lobe, and left caudate nucleus [51]. Connectivity is believed to be reflective of the underlying neuronal structure, so increases in connectivity may be due to the greater white matter organization that is present in BF preterm infants [57]. This effect of human milk on the structural and functional neurodevelopment of preterm infants has been shown to be dosage-dependent, with infants who were BF for a longer duration having improved white matter microstructure [49] and cortical maturation patterns that more closely resemble term-born infants [52].

#### 4.2.3. Supplemented Formula’s Effect on Term Infants

The importance of nutrient composition is emphasized by studies comparing formulas enhanced with additional nutrients to standard formulas. In term infants, a bovine-based formula enhanced with PUFAs and additional vitamins resulted in higher myelination at 6, 12, 18, and 24 months than infants given a control formula that was in line with the recommended nutritional guidelines at that time (2015) [56]. Infants given formula with additional sphingomyelin (SM) were also found to have increases in myelination throughout the brain [54]. Both PUFAs and SM are abundant in human milk, and while commercial formula is often supplemented with both, formula often contains a depreciated composition in comparison [81,82]. These articles suggest that in the absence of human milk, formulas supplemented with additional PUFAs and lipids result in better maturation of neuronal structure.

#### 4.2.4. Supplemented Formula’s Effect on Preterm Infants

This effect is reported in preterm populations as well; Strømmen et al. [44] found that formula supplemented with more calories, amino acids, fatty acids, and lipids had improved maturation in white matter tracts as evidenced by a lower mean diffusivity in cerebral white matter regions. However, this study has relatively few participants (n = 14 enhanced formula, n = 11 control/standard formula) because of the identified increased risk of septicemia in the enhanced formula group. A paper using this same cohort found that infants receiving the enhanced formula experienced an increased rate of electrolyte imbalances (hypophosphatemia, hypokalemia, and hypercalcemia) that may have resulted in higher rates of septicemia [83]. This enhanced formula was within the upper limits of dietary recommendations for preterm infants and highlights the need for more research in animal models to further guide these dietary recommendations.

#### 4.2.5. Effect of Macronutrient Intake on Preterm Infants

The average intake of human milk per diem and macronutrient composition was not reported to have an effect on brain volumes as measured by MRI in preterm infants [45]. This result is supported by the work of Power et al. [48] that did not find an effect of protein, fat, or carbohydrate content in formula on brain volumes. However, both studies have two similar methodological constraints: (1) a moderate degree of homogeneity in the macronutrient composition of diets and (2) the majority of infants did not receive adequate protein as defined by current recommended nutrition guidelines [84]. Several other studies in preterm infants have found that increases in macronutrients such as proteins and fats result in brain structure (MRI) more similar to term-born infants [46] with increases in regional brain volumes [47,50]. MRIs in all these studies were conducted within 1 month of TEA, so discrepancies are unlikely due to the timing of the MRI and instead may be due to differences in macronutrient composition between studies, particularly in protein levels.

## 5. Limitations and Strengths

This review contains a thorough search of the intersection between early infant diet and neuroimaging of both term and preterm infants. By including measures of both structural and functional connectivity, a stronger conclusion can be drawn. An inherent limitation of this review and the papers included is the lack of research into the effect of how complementary feeding during infancy and regular diet during early childhood may contribute to neurodevelopment. If human milk and formula have differential effects on neurodevelopment resulting from the differences in fatty acids and macronutrient compositions, it follows that infants who have their diet supplemented with additional food should also exhibit this effect. We identified no study that followed infants longitudinally to timepoints reasonably associated with beginning complementary foods that presented data on how the intake of these foods may be affecting neurodevelopment. The EEG studies included in this review were largely constrained to one study population, the Beginnings Study, so results summarizing diet’s effect on functional neuronal development as a whole should be cautiously interpreted.

## 6. Summary and Conclusions

While the majority of studies conducted using MRI strongly suggest that human milk is optimal for the neurodevelopment and maturation of both term and preterm infants, more research is needed to support this claim. There remains a substantial gap in understanding how early infant diet effects the brain’s functional development, and more studies using functional modalities such as EEG, MEG, and fMRI in more diverse populations are needed to better address this gap in the literature. Overall, human milk-fed infants tend to have stronger markers of neuromaturation than their formula-fed counterparts; however, this effect is somewhat mitigated by supplementation of formula with fatty acids and other macronutrients in formula-fed infants. These differences in term infants are more subtle than in preterm infants which may be due to the greater nutritional needs of preterm infants [85].

## Figures and Tables

**Figure 1 nutrients-16-01703-f001:**
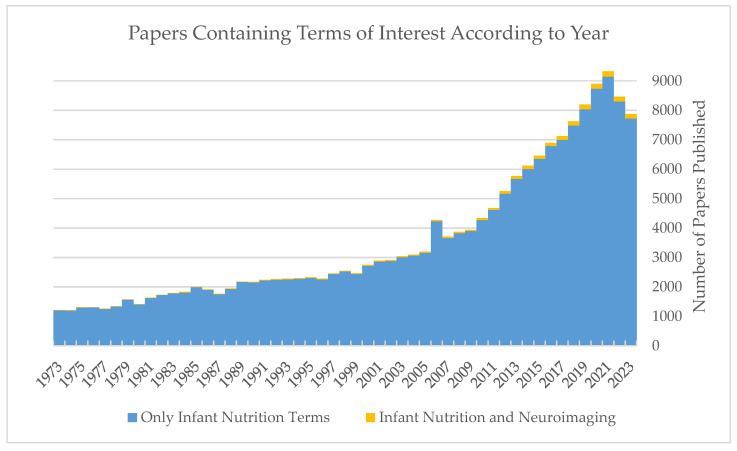
Papers published by year using the nutritional terms identified for this review separately (blue) and in conjunction with the neuroimaging terms (yellow). The last 50 years of publications are included (1973–2023), and full MeSH terms used in the search can be found in Appendix A.

**Figure 2 nutrients-16-01703-f002:**
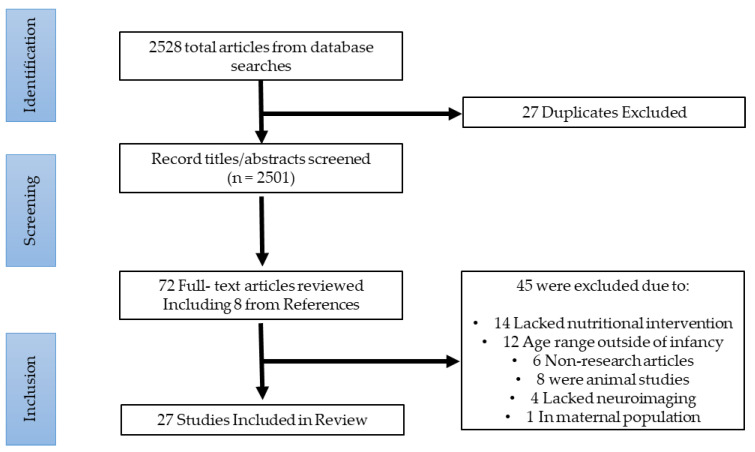
PRISMA flow diagram detailing studies selected.

**Table 1 nutrients-16-01703-t001:** PICOS table summarizing inclusion and exclusion criteria as they pertain to this review.

	Inclusion	Exclusions	Comments
**Population**	Healthy infants and young children up to 2 years of age	Developmental disorders, childhood diseases, nutritional deficiencies, animal studies	Neuroimaging in older populations (2+ years of age) with a retrospective on infant diet were excluded
**Intervention**	Various diets including supplementation	Diets specifically combating a nutritional deficit inherent in the study population	Studies that supplemented either human milk or formula with compounds identified to be essential for neurodevelopment were included as well
**Comparator**	Any	No comparisons made	Any comparisons between different healthy diets were included
**Outcome**	Any outcome that can be quantitatively measured by any neuroimaging technique	Purely psychological or behavioral results	Psychological and behavioral outcomes have been assessed previously [1]
**Studies**	Original quantitative studies	Reviews, abstracts, expert opinions, letters to the editor	

**Table 2 nutrients-16-01703-t002:** Studies included in review. *MRI* magnetic resonance imaging, *sMRI* structural MRI, *dMRI* diffusion MRI, *ACT* anatomically-constrained tractography, *DTI* diffusion tensor imaging, *ASL* arterial spin labeling, *rCBF* relative cerebral blood flow, *EEG* electroencephalography, *TEA* term equivalent age, *FA* fractional anisotropy, *vFM* mcDESPOT-derived myelin water fraction, *MWF* myeline water fraction, *CAVT* cerebral arterial vessel tortuosity, *ERP* evoked related potential, *DHA* docosahexaenoic acid, *AA* arachidonic acid, *SM* sphingomyelin, *PUFA* polyunsaturated fatty acid, *BF* human-milk-fed, *FF* formula-fed, *SF* soy-based formula-fed, *MF* bovine-based formula-fed, *PLIC* posterior limb of the internal capsule.

Study	Publication Year	Modality	Age Group	Number of Subjects per Diet	Major Reported Finding	Comments
Bouglé D [32]	1999	EEG, Auditory and visual ERPs	Preterm, EEG at TEA	15 BF, 14 Formula with LCPUFAs, 11 Formula with short-chain PUFAs	No main effects of diet for either age.	N/A
Pivik RT [34]	2007	EEG, Language ERP	3 and 6 months	15 BF, 18 MF	No main effects of diet for either age.	Beginnings Study
Jing H [35]	2007	EEG, Language ERP	3 and 6 months	20 BF, 21 MF	No main effects of diet for either age.	Beginnings Study
Henriksen C [33]	2008	EEG, ERP related to memory	Preterm, EEG at 6 months	68 Intervention (DHA and AA), 73 Controls	Infants in the intervention cohort had more negative amplitudes to repetitions of a standard image.	Both cohorts received 0.5 mL of study oil per 100 mL of human milk
Li J [36]	2010	EEG, Language ERP	3 and 6 months	40 BF, 51 MF, 39 SF	P350 amplitude: BF > FF at 3 months. N250 and P350 latencies: BF > SF	Beginnings Study
Jing H [37]	2010	EEG, Resting State	3, 6, 9, and 12 months	40 each BF, MF, SF	0–3 Hz: FF > BF at 6 months, BF > FF at 9 months, 3–6 Hz: FF > BF at 6 months, 6–9 Hz: MF > BF at 3 months, MF > SF at 6 months; 12–30 Hz: BF > SF and MF > SF.	Beginnings Study
Pivik RT [38]	2011	EEG, Language ERP	3 and 6 months	75 BF, 88 MF, 76 SF	P350 amplitude: BF < SF to the standard syllable across sites at 6 months.	Beginnings Study
Pivik RT [39]	2016	EEG, Language ERP	4 and 5 months	36 BF, 31 MF, 35 SF	P170 Amplitude at 5 months: BF > SF for deviant stimulus; P350 Amplitude: SF > BF for deviant syllable, BF > SF for standard syllable at 4 months, SF < BF deviant and BF < SF standard at 5 months; P600 Amplitude: MF>SF for standard syllable at 4 months.	Beginnings Study
Pivik RT [40]	2019	EEG, Resting State	6 months	170 BF, 186 MF, 162 SF	Differences in gamma power (BF > SF and BF > MF) in two left-sided regions of the brain.	Beginnings Study
Alatorre-Cruz C [41]	2023	EEG, Language ERP	3, 6, 9, 12, 24 months	127 BF, 121 MF, 116 SF	Differences in P2 latency but not amplitude at 12 months (BF, MF > SF) at frontal left ROI and (SF > MF) at temporal right ROI.	Beginnings Study
Gilbreath D [42]	2023	EEG, Resting State	2–6 months	~100 BF, MF, and SF	Global beta and gamma were increased in BF vs. SF at 2 and 6 months, reflected in source modeling of frontal lobe.	Beginnings Study
Niu W [43]	2020	fMRI, Global Efficiency	Preterm, 40 weeks	30 BF, 20 FF	BF infants exhibited greater global efficiency in comparison to FF.	N/A
Deoni SC [53]	2013	MRI (mcDESPOT) VFm	10 months to 4 years	85 BF, 38 FF, 51 combined BF and FF	Exclusively BF infants had greater VFm in the frontal regions of the brain, formula-fed groups had increased VFm in right optic radiation and occipital lobe.	N/A
Strømmen K [44]	2015	MRI, DTI	Preterm, MRI at TEA	14 Enhanced Nutrition (more calories, amino acids, lipids, fatty acids, and vitamin A), 11 Controls	Enhanced nutrition groups had lower MD values in the cingulum, corticospinal tract, superior longitudinal fasciculi, and uncinate fasciculi.	A significantly higher occurrence of late-onset septicemia was observed in the intervention group
Vasu V [45]	2014	MRI (Volumes), CAVT	Preterm, MRI at TEA	MRI: 19 preterm, 19 term; CAVT 20 preterm, 13 term	Total human milk intake did not influence brain volumes, but did have a positive correlation with CAVT score.	Macronutrient and human milk intake were calculated through medical records
Beauport L [46]	2017	MRI (Lesions)	Preterm, MRI at TEA	42 infants, diets assessed by specific nutrient contents	Increased calories and lipids during the first 2 weeks of life resulted in a reduced risk of a severely abnormal MRI.	Macronutrient and human milk intake were calculated through medical records
Coviello C [47]	2018	MRI (Volumes), DTI (Microstructure)	Preterm, MRI at TEA	103, grouped by protein, fat, and caloric intake	Protein, fat, and calorie intake were positively correlated with cerebellar volume. Calorie, protein, and fat intakes were positively associated with FA in the PLIC.	Macronutrient and human milk intake were calculated through medical records
Deoni S [3]	2018	MRI, mcDESPOT (Myelination)	3 months to 5 years	62 BF, 88 FF (21 A, 28 B, and 39 C)	FF groups had an increased MWF before 1 year, a slower MWF between 1 and 2 years compared to BF.	BAMBAM study. Formulas B and C had higher DHA, ARA, choline, and sphingolipids than Formula A.
Power V [48]	2019	MRI (Volumes)	Preterm, MRI at TEA	81, diets were assessed by protein, fat, and carbohydrate intake	No relationship between nutrition intake and brain volumes.	90% of infants were in line with carbohydrate and fat recommendations; only 3.4% were for protein
Blesa M [49]	2019	sMRI/dMRI; ACT	Preterm, MRI at TEA	27 BF > 75% of time pre-study, 20 BF < 75% of time pre-study	Infants who were BF for longer had increased FA-weighted connectivity and FA in white matter tracts. No differences in global networks or brain volumes.	N/A
Schneider N [54]	2019	MRI (Volumes), mcDESPOT (Myelination)	Birth to 2 years	39 Product A, 28 Product B, 21 Product C	At 1–12 Months: No significant differences. At 12–24 months: Higher SM is associated with more myelin content in the bilateral cerebellum, occipital lobe, visual cortex, internal capsule, parietal lobe, and motor cortices.	BAMBAM study. Minimum time on diet is 3 months, no maximum or total time on diet provided
Ottolini KM [57]	2020	MRI (Volumes), DTI (Microstructure)	Preterm, MRI at TEA	44 BF, 24 FF	BF infants had larger total brain volumes, regional brain volumes (amygdala-hippocampus and cerebellum), and greater regional white matter microstructure organization in the corpus callosum, internal capsule, and cerebellum.	N/A
Hortensius LM [50]	2021	MRI, DTI (Microstructure)	Preterm, DTI at TEA	62 cohort A, 61 cohort B (higher protein and calories)	Cohort B had higher FA multiple white matter tracts; this effect is most associated with the increase in protein (FA was not associated with lipids or calories).	N/A
Berger P [55]	2022	MRI, ASL (rCBF), DTI (Microstructure)	MRI at 1 month	20 mother-infant dyads	Differences in HMO exposure resulted in differences in FA, MD, and rCBF. Certain HMOs are more associated with optimal white matter development.	Human milk from each mother was analyzed for concentrations of candidate HMOs
Zhang Y [51]	2022	MRI (Volumes) and fMRI	Preterm, MRI at TEA	34 BF, 22 FF	BF infants had increased regional gray matter development and function compared with FF infants.	N/A
Sullivan G [52]	2023	MRI (Volumes), DTI (Microstructure)	Preterm, MRI at TEA	67 BF > 75% of time, 68 BF < 75% of time, 77 term-born infants (controls)	Infants who were BF for longer had lower relative cortical gray matter volume and higher mean cortical FA than infants who were BF a shorter time, and had similar FA to controls.	N/A
Schneider N [56]	2023	MRI (Volumes), mcDESPOT (Myelination), DTI (Microstructure)	Birth to 2 years	108 BF, 42 Investigational (increased DHA, AA, B12, folic acid, iron, and SM), 39 Control	Higher myelination was observed in the investigational compared to the control group at 6, 12, 18, and 24 months, higher gray matter volume at 24 months, no differences at any age for WM volumes.	Full comparisons to the BF reference group were not reported

**Table 3 nutrients-16-01703-t003:** Quality Criteria Checklist (QCC; risk of bias) assessment of studies included in this review. Positive (+); neutral (Ø).

First Author, Year of Publication	Primary Research QCC	1. Was the Research Question Clearly Stated?	2. Was the Selection of Study Subjects Bias-Free?	3. Were the Study Groups Comparable?	4. Was Method of Handling Withdrawals Described?	5. Was Blinding Used to Prevent Introduction of Bias?	6. Were Intervention Factors and Any Comparison(s) Described?	7. Were Outcomes Clearly Defined and the Measurements Valid?	8. Was the Statistical Analysis Appropriate?	9. Were Conclusions Supported by Results Considering Biases and Limitations?	10. Is Bias Due to Study’s Funding Unlikely?	Overall Quality
Bouglé D, 1999 [32]		Y	Y	Y	Y	Y	Y	Y	Y	Y	NA	(+)
Pivik RT, 2007 [34]		Y	Y	Y	Y	Y	Y	Y	N	N	Y	(Ø)
Jing H, 2007 [35]		Y	Y	Y	N	Y	Y	Y	Y	Y	Y	(+)
Henriksen C, 2008 [33]		Y	Y	Y	Y	Y	Y	Y	N	Y	Y	(+)
Li J, 2010 [36]		Y	Y	Y	N	Y	Y	N	Y	Y	Y	(Ø)
Jing H, 2010 [37]		Y	Y	Y	N	Y	Y	Y	Y	Y	Y	(+)
Pivik RT, 2011 [38]		Y	Y	Y	N	Y	Y	Y	N	Y	Y	(+)
Pivik RT, 2016 [39]		Y	Y	Y	N	Y	Y	Y	N	Y	Y	(+)
Pivik RT, 2019 [40]		Y	Y	Y	Y	Y	Y	Y	Y	Y	Y	(+)
Alatorre-Cruz C, 2023 [41]		Y	Y	Y	Y	Y	Y	Y	Y	Y	Y	(+)
Gilbreath D, 2023 [42]		Y	Y	Y	Y	Y	Y	Y	Y	Y	Y	(+)
Niu W, 2020 [43]		Y	Y	Y	Y	Y	Y	Y	Y	Y	Y	(+)
Deoni SC, 2013 [53]		Y	Y	Y	Y	Y	Y	Y	Y	Y	Y	(+)
StØmmen K, 2015 [44]		Y	Y	Y	Y	Y	Y	N	Y	N	NA	(Ø)
Vasu V, 2014 [45]		Y	Y	Y	Y	Y	Y	Y	Y	Y	Y	(+)
Beauport L, 2017 [46]		Y	Y	Y	Y	Y	Y	Y	Y	Y	NA	(+)
Coviello C, 2018 [47]		Y	Y	Y	Y	Y	Y	Y	Y	Y	Y	(+)
Deoni S, 2018 [3]		Y	Y	Y	Y	Y	Y	Y	Y	Y	Y	(+)
Power V, 2019 [48]		Y	Y	Y	Y	Y	Y	Y	Y	Y	Y	(+)
Blesa M, 2019 [49]		Y	Y	Y	N	Y	Y	Y	Y	Y	Y	(+)
Schneider N, 2019 [54]		Y	Y	Y	Y	Y	Y	Y	Y	Y	N	(+)
Ottolini KM, 2020 [57]		Y	Y	Y	Y	Y	Y	Y	Y	Y	Y	(+)
Hortensius LM, 2021 [50]		Y	Y	Y	Y	Y	Y	Y	Y	Y	Y	(+)
Berger P, 2022 [55]		Y	Y	Y	Y	Y	Y	Y	Y	Y	Y	(+)
Zhang Y, 2022 [51]		Y	Y	Y	Y	Y	Y	Y	Y	Y	Y	(+)
Sullivan G, 2023 [52]		Y	Y	Y	Y	Y	Y	Y	Y	Y	Y	(+)
Schneider N, 2023 [56]		Y	Y	Y	Y	Y	Y	Y	Y	Y	N	(+)

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
