# Peer review of "A Systematic Review over the Effect of Early Infant Diet on Neurodevelopment: Insights from Neuroimaging"

_nutrients, 2024, doi:10.3390/nu16111703_

Round 1
Reviewer 1 Report
Comments and Suggestions for Authors
General:
1) This article provided a systematic review of a heterogenous set of 26 published studies that met inclusion criteria selected from a large search database to focus on studies of healthy term and preterm infants that included some information regarding type and/or duration of milk feeding in the first several months of life along with some neuroimaging information (broadly defined to include EEG and language ERP in some studies and MRIs in others) in the first year of life.
2) The conclusions regarding effect of various types of feedings on either MRI or EEG or ERP are limited by heterogeneous study designs, inconsistent results among studies, lack of information concerning complementary feeding and lack of longitudinal data.
Introduction:
1) This study attempts to fill in a gap left by other studies which demonstrate the benefits of human milk feeding on cognition and neuromaturation, but which do not include information concerning actual structure and function assessments of the developing brain. Hence this review selected studies that include neuroimaging, broadly defined, in order to fill in that gap.
Methods:
1) Search strategy is described here with results further portrayed in Figure 2 in results.
2) Inclusion criteria indicate the review is restricted to studies of healthy infants (term or preterm) and excluded those with infants with nutritional deficiencies or developmental disorders. Criteria were outlined in a PICOS format in Table 1. The studies had to include some information about infant diets, but it is not clear what minimum of information concerning nature and duration of diet was required or whether any consistent source of dietary information was required. The studies had to include some information on neuroimaging, broadly defined to include MRI (volume and/or structure), fMRI, EEG, auditory and virtual ERPs, ERP related to memory and language; but studies were not restricted as to timing and no study included both MRI and EEG/ERP.
3) No minimum sample size was specified, and multiple studies had <=-50 subjects.
4) No study included any clinical neurological or developmental exams.
5) Some information describing the “neuroimaging” measures utilized was introduced in the Discussion, and would better have been included in the Methods.
Results:
1) The results are largely focused on results of selection (Figure 2), description of studies in Table 2, and study quality Table 3, all of which provide useful information.
2) Though Table 2 does provide a succinct summary of the major reported findings of each study, the further elaboration of those results is presented in the Discussion. Though Discussion would be an appropriate place for comparing studies, especially in terms of consistent vs. contrasting results, some of the basic description of each would be better suited for the results section.
Discussion:
1) As noted above, some of the information about the neuroimaging measures per se would be better moved to the methods section, especially those that may not be as familiar to the readership.
2) See #2 under results.
3) Thought the authors do point out some of the limitations, more should be said about limitations noted in General #2 above.
Reviewer 2 Report
Comments and Suggestions for Authors
Dear Authors,
I've read with interest your paper. The topic is interesting, the literature search appears exhaustive and the paper is well written and easy to follow.
I've only a couple of minor comments:
Figure 1 would be easier to read if you reverse the order of publication years in the x axis (from 1975 to 2023)
Quality assessment is fine but the method used needs to be described in deeper detail
Please try to rotate and fit table 2 within the page margins, otherwise it's impossible to read it
Reviewer 3 Report
Comments and Suggestions for Authors
The authors address an important area, that will be of interest to many. The main issue with the paper as it stands is that neither the abstract nor the results actually give the results of the analysis. Instead the key results are mixed in with the discussion (which is far too wordy).
ABSTRACT
- Some idea of the key findings from the analysis are needed in the study.
INTRODUCTION
- This is too long and can be shortened. For example, the third& fourth paragraphs could be summarized, and some parts moved to the discussion.
- Human milk cannot have both a “slight cognitive advantage” (lines 33-34) and also “substantially enhance cognition” (line 64). Perhaps the authors should include an effect size.
- Perhaps a better structure for the introduction would be (i) the evidence that diet (i.e. human milk) has cognitive benefits, (ii) which individual nutrients have been shown to have cognitive benefits (or are speculated to have such benefits), (iii) how imaging can be used to examine these effects.
METHODS
- There is no reason that the x-axis for Figure 1 should be reversed. It is confusing and counter intuitive. Similarly, showing both sets of results on the same y-axis is unhelpful given the huge difference is numbers between the two results. Look at this figure, I have no clear idea of the number of studies that include both search domains.
- I am unpersuaded that studies for term and preterm infants should be combined.
- In preterm infants are authors talking about mother’s own milk, or donor milk?
- Preterm infants miss an important period of nutrient transfer from the mother. So, I am concerned that the preterm infant may be deficiency, or have suboptimal amounts of these nutrients. This is a problem as the authors seek to exclude infants with nutritional deficiencies.
RESULTS
- EEG and evoked potentials are surely not “imaging studies”.
- The preponderance of preterm infants in the MRI studies is a concern. This is a highly heterogeneous group, with multiple confounders to the provision of mothers own milk, and vary variable comparison diets.
- I don’t know what “otherwise healthy” means for preterm infants, nor how if was defined/ assessed in diet studies.
- Table 2 needs to be divided into studies in preterm and term infants, with all studies for each group grouped together. I think the analysis should also be limited to imaging studies only. More information is needed on the preterm infants in each study as the populations probably vary substantially from study to study in terms of gestational age, birth weight, AGA/ SGA status and comparison diet.
- Table 3 should also be separated in term and preterm studies as for Table 2.
- The results doesn’t include the results I actually was waiting for – what MRI finding where different with HM feeding, whether these findings were consistent from study to study, and consistent between preterm and term infants.
DISCUSSION
- The discussion contains too much explanation of methods used in the studies. This would be reduced by limiting the paper to imaging studies. The summary of finding is too text heavy. Surely, there is some way of summarizing in a table (or 2) what MRI findings changes with HM feeding. This all needs to be in the results, not in the discussion.
-
Round 2
Reviewer 1 Report
Comments and Suggestions for Authors
Authors have adequately addressed the issues/suggestions I raised in my initial review. A couple of minor additional revisions needed:
Line 47: HMO should be abbreviated at the first appearance of human milk oligosaccharids rather than waiting to line 323
Line 148: the total of EEG studies is actually 11 (=2+9), not 10
Author Response
Dear reviewer,
I'd like to thank you again for your helpful contributions and for your thorough observations. I've made both of the minor edits that you suggested, thank you for catching these discrepancies.